# BatCount: A software program to count moving animals

**Ian Bentley[1,2], Vona Kuczynska[3], Valerie M. Eddington[4], Mike Armstrong[5], Laura N. Kloepper[4]***

1 Department of Chemistry and Physics, Saint Mary's College, Notre Name, IN, United States of America, 2 Department of Engineering Physics, Florida Polytechnic University, Lakeland, FL, United States of America, 3 U.S. Fish and Wildlife Service, Missouri Ecological Services Field Office, Columbia, MO, United States of America, 4 Department of Biological Sciences and Center for Acoustics Research and Education, University of New Hampshire, Durham, NH, United States of America, 5 U.S. Fish and Wildlife Service, Kentucky Field Office, Frankfort, KY, United States of America

* laura.kloepper@unh.edu

## Abstract

One of the biggest challenges with species conservation is collecting accurate and efficient information on population sizes, especially from species that are difficult to count. Bats worldwide are declining due to disease, habitat destruction, and climate change, and many species lack reliable population information to guide management decisions. Current approaches for estimating population sizes of bats in densely occupied colonies are time-intensive, may negatively impact the population due to disturbance, and/or have low accuracy. Research-based video tracking options are rarely used by conservation or management agencies for animal counting due to the perceived training and elevated operating costs. In this paper, we present BatCount, a free software program created in direct consultation with end-users designed to automatically count bats emerging from cave roosts (historical populations 20,000–250,000) with a streamlined and user-friendly interface. We report on the software package and provide performance metrics for different recording habitat conditions. Our analysis demonstrates that BatCount is an efficient and reliable option for counting bats in flight, with performance hundreds of times faster than manual counting, and has important implications for range- and species-wide population monitoring. Furthermore, this software can be extended to count any organisms moving across a camera including birds, mammals, fish or insects.

## Introduction

Effective species management and conservation hinges on accurate population information. For species that are cryptic and/or difficult to count, such as bats, traditional population estimates including visual, photographic counts, or mark/recapture techniques are prone to multiple sources of bias [1]. Furthermore, many methods to estimate populations require observers to enter caves or roosts, disturbing threatened and endangered species during sensitive time periods that may cause bats to abandon their roost, such as during the maternity season when

**Data Availability Statement:** All relevant data are contained within the manuscript and its Supporting Information Files

**Funding:** This work was supported by a grant from the National Science Foundation, nsf.gov (Award

Number 1916850), awarded to I.B. and L.N.K., and additional funding from the U.S. Fish and Wildlife Service (fws.gov), the Kentucky Natural Lands Trust (knlt.org), and the Nature Conservancy (nature.org), awarded to L.N.K. The funders had no role in study design, data collection and analysis, decision to publish, or preparation of the manuscript.

**Competing interests:** The authors have declared that no competing interests exist.

adults care for young or during hibernation. Additionally, entering caves can potentially expose colonies to the pathogenic fungus responsible for white-nose syndrome [2, 3] or other zoonotic diseases that humans may spread amongst their populations. Due to these limitations, populations of most major bat caves are monitored less than would be desired to establish presence/absence at roosts, calculate population trends over time, or gain additional life history information on the timing and duration of seasonal migrations. As a result, we lack fundamental information on the population of many bat species worldwide, especially species that are currently listed as threatened or endangered.

In the past several years, advances in technology have made thermal video systems more user friendly and affordable, and many researchers and governmental agencies now use these cameras to record animals in the darkness. Over a decade ago, the U.S. Army Corps of Engineers created proprietary software ("T³"), which was integrated into a camera system to count bats from thermal imagery [4]. However, the software has not been maintained and cannot be used with current thermal imaging cameras. Recent advances in machine learning approaches and image analysis toolboxes have resulted in several algorithms for tracking the movements of animals [5–10], yet these products have not been widely used by users outside of academia, largely due to the perceived training required to run the software; as a result, the few thermal imagery population estimations conducted by biologists outside of academic institutions are often achieved with manual counts of video samples, which is a time-intensive process [M. Armstrong, personal communication; V. Kuczynska, personal communication; N. Sharp, personal communication].

Motivated by the desire for a counting program that is user friendly, requires little training, is free, and can be integrated with video formats from different camera manufacturers and models including standard video, infrared, and thermal imagery, we developed BatCount software. This software was designed in collaboration with U.S. Fish and Wildlife Service biologists, with a goal of quick adoption among management and conservation agencies worldwide. The objectives of this paper are to describe the hardware requirements and overall process for software operation as well as conduct accuracy performance of the software under different field recording scenarios.

## Materials and methods

### Availability and hardware requirements

BatCount v1.24 was developed using MATLAB R2022a (MATHWORKS, Natick, MA) and runs on Windows (Mac OS version in testing). The software uses a standalone interface that does not require the user to purchase or install MATLAB. Rather, specific MATLAB routines and toolboxes that are needed are automatically installed during the software installation. Minimum hardware requirements to operate the software include 4 GB RAM and 2 GB video card RAM, which are standard on most currently available personal laptops. For optimal performance, we recommend a computer with 24 GB RAM and 4 GB video card RAM. Testing of the software was conducted with a Lenovo ThinkStation P340 SFF vPro Core i7-10700 2.9 GHz desktop with 32 GB RAM. Although the software can be used with any type of video recording, such as standard video, infrared, or thermal, we tested the software from available historic and recently collected recordings acquired with three different thermal cameras: 1) A Viento 320 (Sierra-Olympic, Hood River, Oregon, manufacture date 2015) with 320 x 240 resolution recording at 30 frames per second, 2) A FLIR Scion OTM 266 (Teledyne FLIR, Wilsonville, Oregon, manufacture date 2020) with 640 x 480 resolution recording at 30 frames per second, and 3) a FLIR Photon (FLIR, Wilsonville, Oregon, manufacture date 2008) with 320 x 240 resolution recording at 30 frames per second. The software install file, source code, and user guide

can be downloaded at https://tinyurl.com/batcountsoftware, which is a simplified URL that redirects to: http://sites.saintmarys.edu/~ibentley/imageanalysis/pages/BatCount.html.

## BatCount algorithm and operation

BatCount v1.24 first allows users to upload a video for analysis from its graphical user interface. The program supports videos in multiple formats including .avi, .gif, .mj2, .mov, .mpg, .mp4, and .wmv at any resolution and any frame rate, although we recommend a minimum resolution of 640 x 480 and frame rate of at least 30 frames per second for optimal video quality for thermal bat counting. The program uploads videos and partitions the videos into smaller video segments to improve performance as the video is analyzed. Its interface then allows users to a preview any frame of the selected video, navigate between frames, and edit the image for the preview (e.g., crop, zoom). The user can specify the frame range in which to count bats, the maximum and minimum pixel range in which to consider an object a bat, and the threshold, which determines the detection level in which the software will detect an object against the background. The user can also specify one or multiple regions of interest for tracking, which can be either a rectangle or a polygon with user specified vertices. Additionally, users can choose to ignore all objects that are either lighter or darker than the background, which can aid in ignoring a shadow (created from using infrared illumination) or any auto-generated text or numbers on a screen (such as a video clock or frame number). The final user-specified inputs include preview display settings (frame number, crossing counts, internal counts, and overlay grid) and output video settings (tracks, enter and exit, centroid, and bounding box). An example of the software interface is depicted in Fig 1.

The software operates by detecting moving foreground objects (bats) against a background. To account for motion relative to a static background, we use an adaptive process for background determination by calculating the median value of the local segment of video frames (as discussed in [11]). We also re-calculate the relationship between the background and exiting bats over the video duration because the background color will continually change as a result of dropping temperatures and resulting heat loss from the background surface at sunset. The local segmented video frames are used so that the overall lighting is comparable between the background and the frame of interest. The use of a median value as a background is based on the reasoning that if bats are present at any given pixel for fewer than half of the frames, then the median value will contain only background.

The tracking phase of the software results in a count of bats moving across the user specified regions. The software determines connecting lines ("tracks") relating the center of one detected object across subsequent frames using a nearest neighbor approach. More specifically, the tracks are calculated by comparing three sequential frames. First, the center of a bat is determined in the current frame and the prior frame. Based on these positions the center is predicted for where a bat should be on the future frame, assuming linear motion. If the predicted location is within the bounding box for a bat in the future frame, then a line is drawn indicating a correctly predicted future track. The same process is run backward to determine prior tracks. The corresponding tracks for forward and backward tracks are used to determine if a bat has entered or exited a user specified region of interest. These crossing counts are ultimately used to determine overall counts for the videos.

## BatCount output and interpretation

Upon completion of the counting, the software outputs four files: 1) an output summary table, 2) an output settings file, 3) a detailed counting log of the number of bats both in the entire frame and in the region of interest, and 4) if specified by the user, an output video

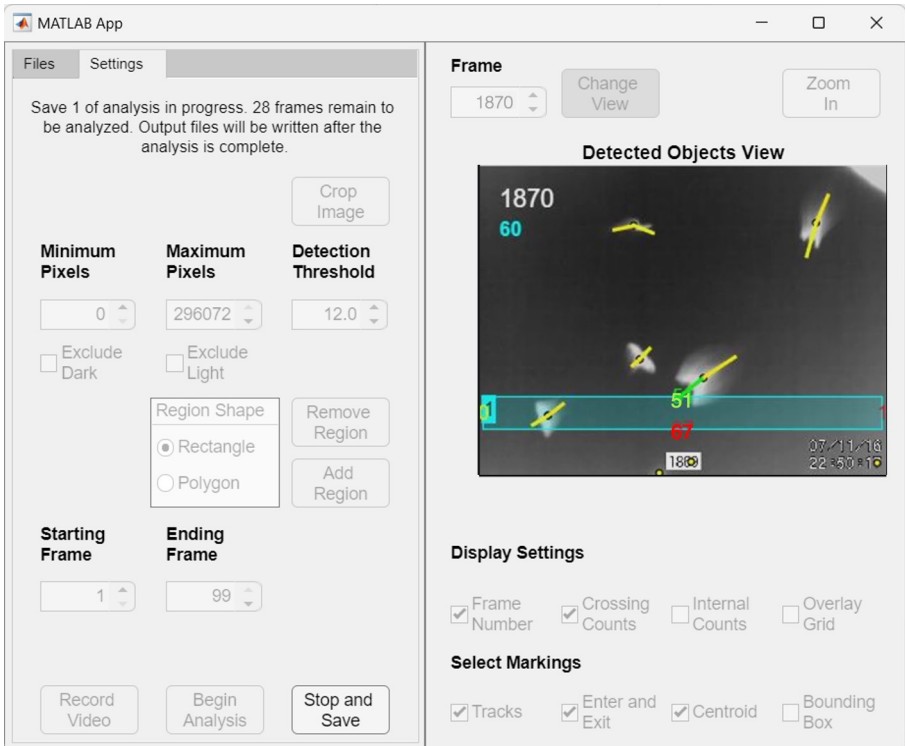

**Fig 1. BatCount user interface with a video loaded and the detected object's view toggled on.** This image was taken during an analysis of a video, so many of the adjustable user parameters appear grayed out and not editable at this stage. A rectangular region of interest has been specified on this frame to count the number of bats that pass through it. The bats' overall flight trajectory starts from the top of the image and continues toward the bottom portion of the screen, intersecting the rectangular region along their path. The frame number 1870 is shown in white, and the crossing sum (60 in this case), which calculates the bats that move through the rectangular region, is displayed below it. A net count of 51 bats have entered the top of the region (shown in green), 67 have left the bottom of the region (shown in red), one net bat has exited the right (shown in red) and no net bats have exited the left side (note the blue highlighted 1, which indicates the number of the selection box, slightly obscures the yellow 0 below). See S1 Video for the original video file used for this analysis, S1 Table for corresponding summary output table and S2 Video for the software output video file. Note: for ease of visibility in the manuscript we electronically manipulated the contrast of the box counting numbers due to partial occlusion by the box and tracking line.

overlaid with detected objects and tracks. An example of an output summary table is shown in Table 1 with the corresponding explanation of the output results table illustrated in Fig 2. See supplemental information for example test video (S1 Video) and corresponding output files (S1 Table and S2 Video).

The software compiles the enter and exit values as bats cross each region of the rectangular box or polygon, as well as calculates two summation metrics. The crossing summation metric, $C_{sum}$, sums the number of bats if bats are moving across the field of view of the camera in one generally polarized direction, such as bats emerging from a cave opening. For a rectangular region of interest this is calculated by summing the larger of the entering or exiting values on

**Table 1. Example output summary table.** See Eqs 1 and 2 for explanation of the crossing and emergence sums. The values represented in parentheses are the actual values calculated for the example shown in Fig 1 (see also S1 Video).

|  | Top | Right | Bottom | Left | Crossing and Emergence Sums |
|---|---|---|---|---|---|
| **Enter** | $T_{enter}$ (88) | $R_{enter}$ (0) | $B_{enter}$ (33) | $L_{enter}$ (0) | $C_{sum}$ (60) |
| **Exit** | $T_{exit}$ (37) | $R_{exit}$ (1) | $B_{exit}$ (100) | $L_{exit}$ (0) | $E_{sum}$ (17) |

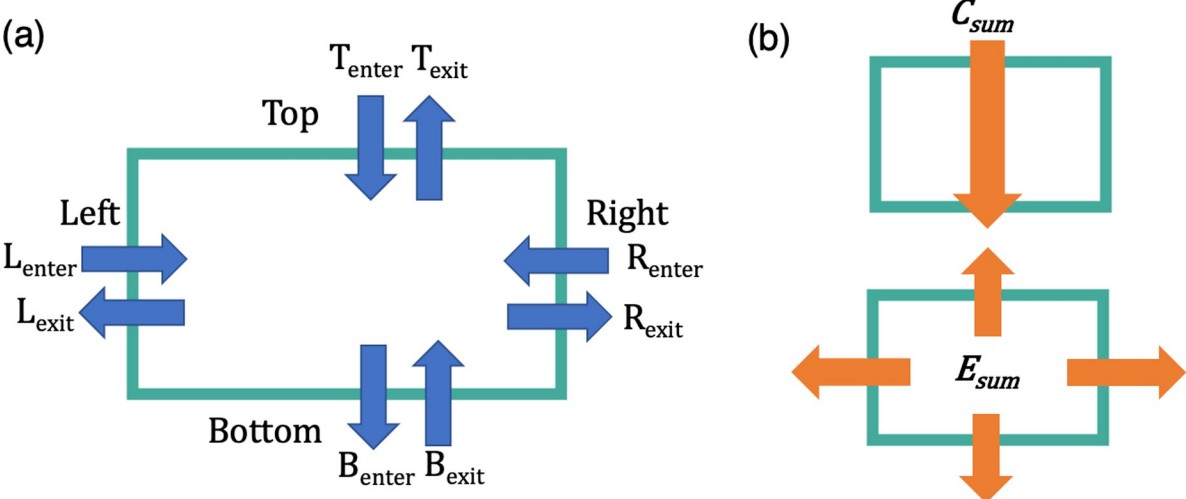

**Fig 2. Illustration of the output summary results table based on the user selected region.** This example shows the output for a rectangular box. (a) The software counts the total number of bats entering and exiting each side of the selection box for the entire video. (b) Illustration of the bat flight profiles that would be appropriate for using the Crossing sum, $C_{sum}$ (Eq 1), and Emergence sum, $E_{sum}$ (Eq 2). The $C_{sum}$ should be used when counting bats transiting across the user selected region, whereas the $E_{sum}$ should be used when counting bats emerging from a central position within selected region.

each side:

$$C_{sum} = \frac{1}{2}\left(\left|(T_> - T_<) + (B_> - B_<) + (L_> - L_<)(R_> - R_<)\right|\right) \tag{1}$$

where T denotes the top side, B denotes the bottom, L denotes the left, and R denotes the right (Fig 2). Here the greater than and less than correspond to the greater or and lesser, values of the entering count and the exiting counts. This automatic determination of the largest value, between enter and exit counts, allows for counting of bats crossing the camera's field of view in any direction. In the crossing sum, the values are divided by 2 to account for the double counting of the same bat entering a region of interest on one side and exiting on another, such as a bat moving from left to right or top to bottom.

The emergence summation metric, $E_{sum}$, corresponds to the number of bats leaving or entering a region of interest, such as if bats are emerging from a bat box, tree, pit cave, or if the camera was pointed directly facing a cave opening. This is calculated by:

$$E_{sum} = \left|(T_{enter} - T_{exit}) + (B_{enter} - B_{exit}) + (L_{enter} - L_{exit})(R_{enter} - R_{exit})\right| \tag{2}$$

where the difference in respective number of bats entering and exiting each side is calculated.

For videos where there is bulk movement across the region of interest the $C_{sum}$ metric is greater and for videos where there is bulk movement into or out of a region of interest the $E_{sum}$ metric is larger. Both output counts are saved in the output data file and the larger of the two values is displayed in the interface below the frame number. For example, Table 1 depicts the actual counts in the summary output file for the example illustrated in Fig 1. The value listed at the top of the selection box in Fig 1 $T_{enter} - T_{exit} = 51$, corresponds to 51 more bats entering (green) the top than had exited. Similarly, the net value $R_{enter} - R_{exit} = -1$, corresponds to one more bat exiting (red) the right then had entered. Similarly, $L_{enter} - L_{exit} = 0$ corresponds to no net bats traveling across the left portion, and $B_{enter} - B_{exit} = -67$, corresponds to 67 more bats exiting the bottom than entering. Based on these count differences: $C_{sum} = 59.5$, which has been rounded to 60 and is displayed below the frame number and written in

cyan to match the cyan color region of interest. $E_{sum}$ = 17, and while displayed in the summary output table is not visible on the software interface because it is the smaller of the two numbers. If $E_{sum}$ was greater than $C_{sum,}$ its value instead would be displayed and shown in cyan. See S1 Video for the original video file used for this analysis, S1 Table for corresponding summary output table and S2 Video for the software output video file.

It is important to emphasize that the user should think carefully about the count values most appropriate for their video. For example, $C_{sum}$ was designed for videos in which bats are truly crossing opposing regions of the selection box, i.e., top to bottom or left to right. For some recording scenarios, bats may be entering crossing adjacent corners, such as entering from the top and exiting the right. In these situations, relying on $C_{sum}$ will substantially under-count the bats, and it would instead be better to use the enter and exit counts from one region of the selection box, such as the top. As such, users of the software should always preview the emergence video to determine the summary table output value that is most appropriate given the overall bat flight behavior.

## Software accuracy

We evaluated the accuracy of the software with thermal recordings from eight different locations: six *Myotis grisescens* (MYGR, gray bats) and two *Tadarida brasiliensis* (TABR, Brazilian free-tailed bat) maternity roosts. Date, location, software accuracy, and camera information for each recording is listed in Table 2. We chose videos with different roost types, species, background clutter, bat densities, and emergence profiles to represent the diversity of applications by the end user. Historical populations of the caves have been estimated between 20,000–50,000 (MYGR caves) and 70,000–250,000 bats (TABR caves). The general placement of the cameras recorded bats either transiting directly across the field of view of the camera (recorded either underneath the emerging bats pointing up or from a side profile) or recorded bats flying towards the camera head-on, with the camera pointed directly at the opening of the cave with bats flying up and out from a central position. Due to the length of recordings and density of bats in the videos at the maternity caves, manual counts of the entire video were prohibitive. Instead, we randomly selected 11–20 replicates (see Table 1) of 900-frame video segments (30 seconds) from each emergence recording for manual counting. Counting was conducted by trained technicians unaware of software program results. During the initial training period, the technicians both unknowingly counted the same video segments and had manual counts within 96.5% of each other. After the training period, technicians unknowingly overlapped 10% of their video segments so we could ensure continued accuracy in counting. Manual counts were conducted with a frame-by-frame analysis using the KMPlayer software (version 4.2.2.58) in 50 frame segments. To expedite counting, we manually counted bats entering and exiting one of the four rectangular regions (the same region and side for each video) and compared the performance of the software to the manual counts.

## Results and discussion

Although the BatCount software was developed with 100% accuracy on object simulated videos, in real-world field settings at bat roosts the accuracy ranged from 50.8 to 94.8% (Table 2). Software performance strongly depended on video quality, with the highest performance achieved for videos with strong contrast between the bats and the background, which is achieved both by camera resolution and distance from the camera to the bats, and minimal overlap of bats in the videos. Our peak accuracy of 94.8% is slightly higher than the reported peak accuracy of 93% for the T3 system [4]. Camera model and manufacture date also affected software performance, with videos recorded by the FLIR Scion (manufactured 2020) and

**Table 2. Recording date (month/day/year), location (county, state), camera type, average number of bats per 900-frame segment, number of video segments included in the analysis, and overall software accuracy for each of the eight recordings used to evaluate the software.**

| CaveID | Recording date | Location | Camera | Bats/seg | # video segments | Software accuracy |
|---|---|---|---|---|---|---|
| MYGR1 | 07/17/2020 | Camden, MO | Viento | 128 | 20 | 91.3% |
| MYGR2 | 06/25/2021 | Taney, MO | FLIR scion | 188 | 20 | 90.1% |
| MYGR3 | 06/25/2021 | Wright, MO | FLIR scion | 163 | 19 | 94.8% |
| MYGR4 | 06/22/2021 | Oregon, MO | FLIR scion | 252 | 11 | 72.0% |
| MYGR5 | 2012 | Wilson, TN | FLIR photon | 146 | 20 | 50.8% |
| MYGR6 | 08/13/2021 | Nelson, KY | FLIR photon | 23 | 19 | 56.7% |
| TABR1 | 06/13/2016 | Woods, OK | Viento | 776 | 18 | 83.6% |
| TABR2 | 06/15/2016 | Woodward, OK | Viento | 887 | 15 | 70.8% |

Viento (manufactured 2015) cameras (average performance 85.6 and 81.9%, respectively) out-performing those recorded by the FLIR photon camera (average performance 53.8%, manufactured 2008). The poor accuracy of the videos MYGR5 and MYGR6 was due primarily to a combination of low background contrast and poor video resolution; even our trained technicians struggled to visually discriminate bats against the background. Therefore, we cannot disambiguate whether the poor performance for these two locations is due to camera quality, environmental conditions, or both. Due to these limitations, we removed MYGR5 and MYGR6 from further analysis.

Fig 3 illustrates the accuracy of the software as a function of the number of bats in each 900-frame (30 second) segment for each cave location. At all locations, the software underestimated bat counts. The data are best represented overall with a logarithmic fit, in which accuracy is low at low numbers of bats (< 50 bats per 30 second segment) but remains relatively stable around 83% accuracy for medium densities of bats (between 50 and 800 bats per 30 second segment). When bats began to overlap at higher emergence densities (TABR1, TABR2), the chance of the software counting two bats as one increased, and accuracy begins to slightly decline. We are currently developing a new method to better count overlapping bats and expect an increase in software accuracy with its incorporation. All updates of the software will be released on the software website and announced via authors' social media.

The variation in performance of our software based on camera type and location highlights the importance for continued validation of automated software in real-world scenarios. For example, although lab testing on an "ideal" video yielded 100% accuracy, the range of accuracy of our software at bat cave recordings could result in large fluctuations in overall population counts if an end-user chose to use our software as a black-box tool (i.e. not investigating the performance of the software). For this reason, in scenarios when a highly accurate population estimate is needed, such as for the listing or delisting of endangered species, we recommend that software users obtain a site-specific accuracy estimate. This can be achieved by manually counting a small number of video segments and comparing the software counts to the manual counts. From this, users can determine how much the software underestimates the true population size and adjust population count accordingly. For situations in which manual counts may be prohibitive due to resources, if videos have been recorded with a camera with at least 640 x 480 resolution and 30 frames per second, and the population of the roost is estimated to be less than 500,000 individuals, we recommend the end user assumes the software underestimates the true population by up to 17% (average accuracy of our software across all locations is 83%) and instead provides a range estimate in total population.

Although the exact processing time of the software depends on the length of the video and number of bats in each recording, we can make some general statements about the software

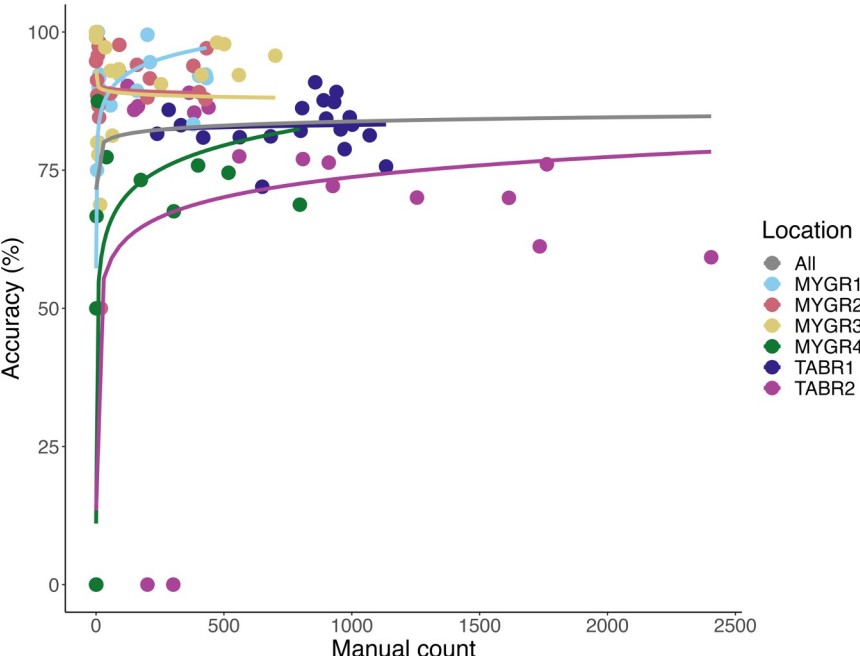

**Fig 3. Performance curves based on number of bats in each video segment.** At low numbers of bats ($<$ 50 bats per 30 second segment), the software demonstrated variable accuracy. At medium numbers of bats (between 50 and 800 bats per 30 second segment), the software performance remained stable, with location affecting overall accuracy. At high numbers of bats ($>$ 800 per 30 second segment), performance began to decline as bats overlapped.

processing time. Using the minimum hardware requirements listed to run the software, the software processes approximately 1 frame per second. Computers with the recommended specifications can process approximately 2 frames per second. For example, an emergence that lasts 60 minutes and was recorded at 30 frames per second would take approximately 15 hours to process using the recommended specifications. This time can be partitioned by counting specific segments of the longer emergence video. We also found it helpful to run the counting software overnight or over a weekend. In comparison, our trained technicians manually counted the more challenging videos at a rate of 1 frame every 2 minutes. Thus, with standard PC equipment our software can count bats 250 times faster than human effort and reduces human bias. The speed of the software can be further accelerated by using a supercomputer, and by using newer PCs with faster processing capabilities.

Due to its intuitive graphical user interface, this software does not require the user to have expertise in any coding languages, and as such is appropriate for broad use among researchers, students, and even the general public. Furthermore, by paring down the output results of the software and including a summary table and output video, we have simplified the results to include the information most relevant to end users. Although created with the main application of counting bats, due to the modifiable input parameters this software can also be used to count birds, mammals, fish or insects. As video technology continues to decrease in cost and increase in resolution, and as computing power continues to evolve, this software can be a powerful, free tool to count animals for conservation, increasing the accessibility of video counting worldwide.

## Conclusion

In conclusion, with our performance testing we know that the current version of our software is highly accurate when recording gray bats with a thermal camera recording at 640 x 480 resolution and 30 frames per second. Future releases of the software will increase performance for dense bat flights. By developing the software in close consultation with and testing from end-users, we have developed a counting software that is intuitive, easy to use, and provides informative summary output including total counts and an output video. This software eliminates the need to exhaust our most precious resource as a conservation community—time. We are currently working with end-users to develop and implement best practices for both placement of cameras in the field and placement of the user-defined selection boxes for software counting. This software provides a free and powerful tool to obtain population counts of bats emerging from roosts and can be a valuable resource to aid in population estimation and species conservation.

## Supporting information

**S1 Video. Example video file.** Original video file used for generating the output displayed in Fig 1 and the corresponding S1 Table output file.
(MP4)

**S2 Video. Example output video.** Software output video based on S1 Video.
(MP4)

**S1 Table. Example output table.** Summary output table for the S1 Video.
(XLSX)

**S2 Table. Data reported in manuscript.** Manual and software counts for all video segments analyzed in the manuscript.
(XLSX)

## Acknowledgments

The authors would like to thank Z-Bar Ranch, Melynda Hickman, Missouri Department of Conservation, Tumbling Creek Cave Foundation, John and Jean Swindell, and Mark and Daniel Mauss for access to field recordings, Jordan Meyer, Dave Woods, Stephanie Dreessen, Cassi Mardis, and Cory Holliday for assistance with video collection, Jim Cooley and the Cave Research Foundation for field assistance, and Lindsey McGovern and Zachary Ahmida for manual counting. We would also like to thank Alexandria Weesner for developing an alternate prototype tracking code.

## Author Contributions

**Conceptualization:** Ian Bentley, Laura N. Kloepper.

**Data curation:** Ian Bentley, Vona Kuczynska, Valerie M. Eddington, Laura N. Kloepper.

**Formal analysis:** Valerie M. Eddington.

**Funding acquisition:** Ian Bentley, Vona Kuczynska, Mike Armstrong, Laura N. Kloepper.

**Investigation:** Vona Kuczynska, Valerie M. Eddington.

**Methodology:** Ian Bentley, Laura N. Kloepper.

**Project administration:** Laura N. Kloepper.

**Resources:** Vona Kuczynska, Mike Armstrong.

**Software:** Ian Bentley.

**Supervision:** Laura N. Kloepper.

**Validation:** Vona Kuczynska, Valerie M. Eddington.

**Visualization:** Valerie M. Eddington, Laura N. Kloepper.

**Writing – original draft:** Ian Bentley, Laura N. Kloepper.

**Writing – review & editing:** Ian Bentley, Vona Kuczynska, Valerie M. Eddington, Mike Armstrong, Laura N. Kloepper.

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
