## [Decision Letter · Decision Letter 0]

10 Jan 2023

PONE-D-22-30737BatCount: A software program to count moving animalsPLOS ONE

Dear Dr. Kloepper,

Thank you for submitting your manuscript to PLOS ONE. After careful consideration, we feel that it has merit but does not fully meet PLOS ONE’s publication criteria as it currently stands. Therefore, we invite you to submit a revised version of the manuscript that addresses the points raised during the review process.

We look forward to receiving your revised manuscript.

Kind regards,

Lalit Kumar Sharma

Academic Editor

PLOS ONE

Journal Requirements:

   "This work was supported by the National Science Foundation (Award Number1916850), the U.S. Fish and Wildlife Service, the Kentucky Natural Lands Trust, and the Nature Conservancy. The findings and conclusions in this article are those of the authors and do not necessarily represent the views of the U.S. Fish and Wildlife Service. The authors would like to thank Z-Bar Ranch, Melynda Hickman, Missouri Department of Conservation, Tumbling Creek Cave Foundation, John and Jean Swindell, and Mark and Daniel Mauss for access for field recordings, Jordan Meyer, Dave Woods, Stephanie Dreessen, Cassi Mardis, and Cory Holliday for assistance with video collection, Jim Cooley and the Cave Research Foundation for field assistance, and Lindsey McGovern and Zachary Ahmida for manual counting. We would also like to thank Alexandria Weesner for developing an alternate prototype tracking code."

  "This work was supported by a grant from the National Science Foundation, nsf.gov (Award Number

1916850), awarded to I.B. and L.N.K., and additional funding from the U.S. Fish and Wildlife Service (fws.gov), the Kentucky Natural Lands Trust (knlt.org), and the Nature Conservancy (nature.org), awarded to L.N.K. The funders had no role in study design, data collection and analysis, decision to publish, or preparation of the manuscript."

Additional Editor Comments:

Please address the minor corrections suggested by one of the reviewers.

Reviewers' comments:

Reviewer's Responses to Questions

**Comments to the Author**

1. Is the manuscript technically sound, and do the data support the conclusions?

Reviewer #1: Yes

Reviewer #2: Yes

2. Has the statistical analysis been performed appropriately and rigorously? 

Reviewer #1: Yes

Reviewer #2: Yes

3. Have the authors made all data underlying the findings in their manuscript fully available?

Reviewer #1: Yes

Reviewer #2: Yes

4. Is the manuscript presented in an intelligible fashion and written in standard English?

Reviewer #1: Yes

Reviewer #2: Yes

5. Review Comments to the Author

Reviewer #1: This is great work delivering an important tool for conservation; Being able to use an system for automated counting, with no or little need to enter roosts, is great in the context of WNS, but also for many different situations where visual counts in roosts or visual counts of bats emerging from roosts, are difficult (e.g. bats roosting in or emerging from tree roosts or complex buildings, dangerous or poorly accessible underground sites).

The text is clear and step by step guiding the reader through both the concept of deducing the relevant info from the frames and turning these into quantitative info, as well as the technical details this involves; this open approach is not only scientific due diligence, but important for future end-users and for learning from their feedback and their experiences.

Validation/calibration is as yet done using a limited set of bat roosts – I would like to stress the importance of more testing with different types of roosts, providing different challenges regarding visibility. Since this is free and open software, open to be used and experimented with, this is no reason to not publish this paper.

Reviewer #2: Dear authors,

I found your manuscript extremely interesting and useful, well-presented, sound and nicely written. Congratulations for an excellent work!

I have attached a new pdf file with some comments and suggestions that I hope may help to improve the clarity of the manuscript and future impact of it on your audience. Most of my comments are related to the applicability of the algortihm in other caves, underground roosts and countries (especially developing countries), as well as regarding the structure of the manuscript (in my opinion some sections are too long to be easily understood).

Apart of that I wanted to congratulate all of you for your hard work and dedication, and for offering such an extraordinary tool, user-friendly and intuitive, for free, to the world.

Looking forward to try it in the field.

6. PLOS authors have the option to publish the peer review history of their article (what does this mean?). If published, this will include your full peer review and any attached files.

Reviewer #1: No

Reviewer #2: No

---

## [Author Response · Author response to Decision Letter 0]

16 Jan 2023

Response to Reviewers

Reviewer #2: (we thank this reviewer for the detailed comments to improve our paper) 

Line 20: And maybe also because researchers assume an elevated cost to be implemented, especially in developing countries? 

If not included here, I would definitely include a mention of the potential positive impacts of this technique in developing countries.

Authors’ Response: This is a great point. We included “and elevated costs” in the abstract and in the discussion include the positive impacts of this technique for increasing accessibility of video counting worldwide.

Line 22: I would specify here the approximate sizes of the colonies for which you have already tested the software. 

This is something really rellevant if you aim to promote its use widely.

Authors’ Response: We included a mention of historical population sizes here. 

Line 22: emerging/from

Authors’ Response: fixed

Line 25: You have not mentioned here the speed improvement in the analyses compared to human-visual counts. It would be interesting to add a short comment about this.

Authors’ Response: good point. We added a phrase to highlight the speed improvement here. 

Line 33: "multiple sources of biases" maybe?

Authors’ Response: added. 

Line 36: " or during hibernation"

Authors’ Response: added. 

Line 37: Since your software will be used worldwide (hopefully) I would not limit or contextualize the whole manuscript in the USA/Canada. Why not adding something like that? "or many other zoonotic diseases that human may spread amongst their populations"

Authors’ Response: This is a great point and highlights our bad judgement in making this US-centric. We have included this statement. We certainly don’t want the application of this software to be limited to the US and Canada!

Line 42: I would not use this sentence here. Although your sampling sites are all based in the US, your experiments have been taken place in American caves, and funding probably come from the US, I understand that the BatCount software has been designed to be used worldwide. I would try to avoid direct references to the US (too specific) and present it as a new method to be used in any country, without biases.

Authors’ Response: Great point. We have removed this statement. 

Line 55: I would cite some examples here (covering several continents if possible).

Authors’ Response: Our information from this statement came from personal communication from our agency reps, who often use the counting information for internal documents or final reports (and so don’t explicitly publish the counts came from manual counting of thermal videos). We included a reference to the personal communication here. 

Line 57: Nowadays there are large differences in terms of costs and availability of thermal cameras and infrared cameras. In many caves in the world, most of the countings seems to be carried out with IR cameras, instead of thermal (because of the price). Therefore, I would recommend to explain very clearly at the beginning whether your software has been designed to be used with thermal, IR or both types of images.

That would probably attract more users :)

Authors’ Response: Great point. We clarified that this can be used with standard video, infrared, and thermal videos.

Line 60: This is a very personal comment, but I do not believe this should be included in the introduction. I would rather place it in the Results/Discussion. 

If you end up removing these sentences, the paragraph should be slightly reshaped to inlude the specific objectives of the paper (right after presenting the overarching goal of developing a user-friendly software).

Authors’ Response: Agree. We made this change and modified the paragraph to include objectives of the paper. 

Line 73: I am not sure if this is possible, but it would be really great to have it running by the date of the manusript acceptance. The paper would gain in force and elegance.

Authors’ Response: We agree, but we need a bit more time (and money!) to get this working, as one of the senior authors is transitioning to a new institution. We hope within a year to have this available for Mac OS users. 

Line 76: I am not sure about this, but, whether most personal laptops already count with these features, it would be great to include a sentence about this, to make sure the reader will be motivated to try it ;)

Authors’ Response: Great tip. We clarified this. 

Line 77: I would also split the sentence here or rephrase it, in order to clarify which requirements are the minimum, and which are the optimal.

Authors’ Response: We split this up, and included the specific computer model we used for testing.

Line 78: Since many bat counts in caves are currently carried out using IR cameras or video cameras with Night-shot vision worldwide, it would be nice to specify here if this software can be used only with thermal or it can also be used with IR.

Authors’ Response: We clarified that this can be used with any type of video recording and clarified why we chose thermal for our testing. 

Line 86: This sub-section is considerably long compared with the other parts within the Materials and Methods section. I would try to split it in 2 or 3 sub-sections to turn the reading easier and friendlier to the non-expert readership.

Authors’ Response: Agree. We split this up into “algorithm and operation” and “output and interpretation”

Line 89: Although it can be used with any resolutiona nd frame rate, I am sure that some minimum requirements are needed. It would be really great if you could add some more info about the video cameras that are necessary to get nice results.

Authors’ Response: Great point--we included recommended parameters for optimal video quality here. 

Line 121: Is there any way to explain how much different the bats (white dots) need to be from the background, to be efficiently detected?

Authors’ Response: Unfortunately, no. We haven’t found a way to quantify this yet. 

Line 214: six/two

Authors’ Response: Changed. 

Line 218: It would be very important to specify the magnitude of these colonies. Are we talking about how many animals in total? I know that you mention the number of animals per second, but I think it would be better to mention approximate colony sizes.

Authors’ Response: Agree. We included the historical population estimates. 

Line 220: It would be great to add the range of real-time recordings that thes 900 frames represent. Assuming that you mostly had 30fps recordings, this would represent 30s of recordings.

Authors’ Response: We included this.

Line 220: how many?

Authors’ Response: We included this. 

Line 225: If there is any specific and detailed reference about this counting protocol (even in a website or similar resource) it would be great to include it here.

Authors’ Response: We don’t have any references for the counting protocol. The technicians just counted bats entering and exiting one of the four rectangular regions, which we explain at the end of the paragraph. 

Line 232: eight

Authors’ Response: Changed

Line 235: I would change the order: from 50.8% to 94.8%

Authors’ Response: Changed

Line 241: This is a very substantial difference in terms of performance. The decrease in accuracy seems rather dramatic, and might eventually discourage people of using your software. You mention here that this could result from the reduced quality of the images. I think it would be necessary to add here some more specific information about the minimum quality/resolution/etc. needed to get the highest accuracies.

Authors’ Response: Agreed. Manufacture date also plays a key role here, and we added the manufacture date of the cameras to show how this likely plays a role in the poor video quality. 

Line 247: According to the accuracies reported here, and considering the total size of the colonies, how big is the error in terms of total estimation of the colonies? 

This information would be very interesting to be added as a complement of the accuracy analyses, and would convince/not convince the audience to use it, if the main aim of the projects is to determine total number of bats in a cave.

Authors’ Response: We agree with this, but this is a bit complicated since even the largest caves have emergence profiles with periods of dense emergence followed by not-so-dense emergence. For example, TABR2 ranges between 200 and 2500 bats per 30-second segment. For the MYGR caves it’s a bit more straightforward since the population count ranges between 1 and 500 bats per 30-second period. Also, error is site/camera specific. We hesitate to use a blanket statement for total estimation of the colonies, as we don’t want someone to blindly use the program and create a population estimate. Rather, our goal of this paper is to emphasize the importance of folks doing some small ground-truthing or error analysis of their own before trusting the “black box” of a program. We do understand, however, that this may be a big time commitment for many. We have expanded our discussion of this to explain that although the software was developed with 100% accuracy on simulated “ideal” videos, in real-world settings the accuracy is highly situation specific and depends on camera quality, density of bats, and distance the camera is from bats (which affects pixel size and ultimately detection). We think a good compromise is to state the overall population estimation accuracy for a recording of a cave with a specific population size and using the optimal camera settings, and we included this in the results and discussion in a paragraph explaining best practices for translating software output to true population count. 

Line 248: If this error is constant in all the locations, do you think there might be a way to automatically compensate it? Is the error constant across locations?

Authors’ Response: We think we explained some best practices for how end-users could incorporate error rates into total population sizes in our revised discussion. 

Line 250: I think it would be great to add some approximate numbers here.

Authors’ Response: done

Line 251: similar comment here. 

Authors’ Response: done

Line 252: slightly decline?

Authors’ Response: changed.

Line 253: Because this is still in development, and it might change in the future, I would recommend to be more vague here. For instance, I would not mention "Neural Network approach", but I would say "a new method to..."

Authors’ Response: changed.

Line 257: Because the accuracy changes substantially in all the locations, would you recommend the users to analyse how good is this software for each of the new locations (new underground roosts) before relying entirely on the automatic detection software? 

I would do so, in order to make sure that all the users are fully aware of the precision of the software in their caves.

Authors’ Response: Yes! We included a new paragraph describing best practices so users don’t view our software as a “black box” with specific accuracy rate.

Line 268: But using the minimum requirements or the optimal ones?

Authors’ Response: optimal. We clarified this.

Line 275: This component here is super-interesting, but remains vaguely explained for those who are not proficient or experts on the topic. I would explain it a little bit more or remove it. Also I would cite other relevant papers to provide some background.

Authors’ Response: Agreed. We removed this statement. 

Line 278: It would be useful to state the minimum resolution for a camera that you would recommend. 

I missed a little bit more information about the thermal cameras, and video resolutions used in the study.

Authors’ Response: We clarified this here. 

Line 284: I kind of missed a paragraph/section explaining the best methods to place your camera to increase the performance of the software. I find it great to mention that here, but it would be great to add this information somewhere.

Authors’ Response: We included a clarification on the placement of the cameras for our testing videos when we introduce them in the methods. We hesitate to make a general recommendation on how to place the camera, as it truly is site-specific. The most important thing is for the best contrast of the cameras and capturing the entire colony.

---

## [Editor Report · Decision Letter 1]

17 Feb 2023

BatCount: A software program to count moving animals

PONE-D-22-30737R1

Dear Dr. Kloepper,

We’re pleased to inform you that your manuscript has been judged scientifically suitable for publication and will be formally accepted for publication once it meets all outstanding technical requirements.

Kind regards,

Lalit Kumar Sharma

Academic Editor

PLOS ONE
---

## [Editor Report · Acceptance letter]

7 Mar 2023

PONE-D-22-30737R1 

BatCount: A software program to count moving animals 

Dear Dr. Kloepper:

I'm pleased to inform you that your manuscript has been deemed suitable for publication in PLOS ONE. Congratulations! Your manuscript is now with our production department. 

Kind regards, 

on behalf of

Dr. Lalit Kumar Sharma 

Academic Editor

PLOS ONE